# Immune-Related Urine Biomarkers for the Diagnosis of Lupus Nephritis

**DOI:** 10.3390/ijms22137143

**Published:** 2021-07-01

**Authors:** María Morell, Francisco Pérez-Cózar, Concepción Marañón

**Affiliations:** GENYO, Centre for Genomics and Oncological Research Pfizer, University of Granada, Andalusian Regional Government, PTS, 18016 Granada, Spain; maria.morell@genyo.es (M.M.); francisco.perez@genyo.es (F.P.-C.)

**Keywords:** Lupus nephritis, urine biomarkers, non-invasive diagnosis, immune effector

## Abstract

The kidney is one of the main organs affected by the autoimmune disease systemic lupus erythematosus. Lupus nephritis (LN) concerns 30–60% of adult SLE patients and it is significantly associated with an increase in the morbidity and mortality. The definitive diagnosis of LN can only be achieved by histological analysis of renal biopsies, but the invasiveness of this technique is an obstacle for early diagnosis of renal involvement and a proper follow-up of LN patients under treatment. The use of urine for the discovery of non-invasive biomarkers for renal disease in SLE patients is an attractive alternative to repeated renal biopsies, as several studies have described surrogate urinary cells or analytes reflecting the inflammatory state of the kidney, and/or the severity of the disease. Herein, we review the main findings in the field of urine immune-related biomarkers for LN patients, and discuss their prognostic and diagnostic value. This manuscript is focused on the complement system, antibodies and autoantibodies, chemokines, cytokines, and leukocytes, as they are the main effectors of LN pathogenesis.

## 1. Introduction

Systemic lupus erythematosus is an autoimmune disease that can affect any organ of the body [1]. The kidney is one of the main organs affected by SLE, and lupus nephritis (LN) is significantly associated with an increase in the morbidity and mortality of these patients [2]. The involvement of the kidney during the course of the disease affects between 30% and 60% of adult SLE patients. Approximately 10–20% of these patients will progress to end-stage renal disease within 5 years after diagnosis, while 40% of them with LN will develop chronic kidney disease [3]. In LN, six histological classes are defined based in microscopic lesions and immune complexes (IC) distribution [4]. Class I LN (Minimal mesangial) represents <20% of all cases of nephrotic syndrome that undergo renal biopsy, and Class II LN (Mesangial proliferative) accounts for 7–22% of all cases, generally presenting with isolated haematuria, low-grade proteinuria, and normal renal function. Class III and IV LN (Focal and Diffuse LN) bear the most severe prognosis and require prompt immunosuppressive treatment [5].

Pathogenesis of LN is mediated by leukocyte infiltration and autoantibody binding to nuclear and non-nuclear autoantigens and/or formation of circulating IC containing autoantibodies, deposed on different parts of the glomeruli [6] (Figure 1). Pro-inflammatory cytokines and effector cells of the immune system lead to an inflammatory organ disease where a variety of cytokines are clearly associated with SLE activity [7]. However, there is still a lack of sufficient knowledge of which immune-pathological pathways are involved [8].

Nowadays, treatment of LN patients usually involves immunosuppressive therapy, typically with mycophenolate mofetil or cyclophosphamide along with glucocorticoids, although these treatments are not uniformly effective [9]. In addition, they are not specific for the disease, and they have shown side effects in patients with renal damage. Moreover, it has been considered that the limit of what conventional drugs can achieve has probably been reached [10]. Kidney biopsy is the gold standard for establishing tissue diagnosis, prognosis, and treatment in LN. However, it is a costly and risky procedure, making it unsuitable for the early detection of renal pathology or to monitor the response to treatment. Currently, laboratory markers for LN such as proteinuria, creatinine clearance, urine protein/creatinine ratio, anti-double strand DNA autoantibodies (anti-dsDNA), and low plasma complement levels (mostly C3 and C4) have been applied in monitoring LN activity in daily clinical routines [11]. Nonetheless, these clinical parameters lack sensitivity and specificity to reflect the real-time renal immunopathological activity and chronic tissue damage [12]. Therefore, novel biomarkers able to discriminate lupus renal activity and its severity, predict renal flares, and monitor treatment response and disease progress are clearly necessary.

Current guidelines for the treatment of LN patients recommend changing the induction therapy if there is no response after 3–6 months [13,14], but unfortunately there is no consensus on the definition of treatment failure [15]. Clinical and histological remission are not always correlated, and SLE patients can be in a stage called silent nephritis, characterized by isolated low-level proteinuria [16] or high anti-Smith (Sm) autoantibodies associated with low complement haemolytic activity [17]. However, the indications of repeated renal biopsies are controversial [18]. An improved renal inflammation monitoring using non-invasive biomarkers could help to identify patients at risk of progression or in therapeutic failure.

In contrast to other biological sample sources, such as serum or tissue, urine sampling is non-invasive, allowing frequent monitoring, and can be self-collected, transported, and stored easily. Furthermore, urinary biomarkers seem to be more promising than serum markers in the study of LN, given that they derive from tissues of the urinary system [19], so that they can reflect its current pathological status [20]. Thus, urine is an attractive source for finding potential biomarkers in the study of LN.

Many immune-related mediators or antibodies may be excreted into urine from the inflamed and damaged kidneys of LN patients (Figure 1). Leukocytes infiltrating the kidney can also be voided and recovered in the urine sediment [21]. In this context, several urinary biomarkers have been evaluated for diagnosis and monitoring of treatment responses in LN patients during the last decades [12]. However, most of these biomarkers have only been tested in cross-sectional studies, and only a few have been evaluated in longitudinal studies and independently validated [12,22]. Evaluation of urine biomarkers in clinical care may predict development of renal flares and determine therapeutic treatment. Here, we summarize the main immune-related urinary biomarkers that have been described as potential non-invasive biomarkers of LN. We focus on the biomarkers directly related with immune effectors mediating LN pathogenesis (antibodies and autoantibodies, complement, cytokines, and leukocytes), as they can better reflect the inflammatory status of the kidney and, consequently, the LN activity (Summarized in Table 1).

## 2. Autoantibodies

SLE is an autoimmune disease characterized by production of autoantibodies against self-molecules present in the nucleus, cytoplasm, and cell surface. The prevalence of serum antinuclear autoantibodies among SLE patients is very frequent, but they may also be detected in patients with other autoimmune diseases and malignant or infectious diseases, as well as healthy controls. Despite this, serum anti-dsDNA antibodies are considered a diagnostic marker and one of the classification criteria for SLE [23]. In the case of LN, while some studies reported that increased titers of serum anti-dsDNA antibodies precede LN flare, others documented that many patients with anti-dsDNA never develop LN, and some patients with LN do not have anti-dsDNA [15].

There is evidence of the presence of IC deposition in renal biopsies of patients with LN [24]. Nevertheless, the mechanisms that lead to the formation of immune deposits and development of renal damage remain unclear [25,26]. IC formation and deposit in the kidneys are most likely involved in the mechanism for urinary excretion of autoantibodies in SLE (Figure 1). However, despite their importance of renal pathology in SLE, and numerous studies of autoantibodies in serum and IC deposits in kidneys [27,28,29,30,31,32], there have been few reports of specific autoantibodies in the urine of SLE patients (Table 1).

Meryhew et al. [33], were the first to report the presence of specific antinuclear antibodies in the urine of SLE patients. They detected anti-Sm, anti-ribonucleoparticles (RNP), anti–Sjögren’s-syndrome-related antigen A autoantibodies (>anti-SSA or Ro), and anti-dsDNA in the urine of these patients, many of them associated with abnormal renal function signs, such as proteinuria. IgG was the most frequent immunoglobulin class present in the urine. A similar observation was made in a study measuring anti-RNA polymerase I (RNAPI) in the urine of SLE patients [34]. A further study [35] also reported the presence of autoantibodies against RNAPI, DNA, SSB (La), and ribosomal P proteins in the urine of SLE patient. In addition, the presence and relative concentration of urinary autoantibodies correlated with disease activity. It is worth mentioning that in many cases specific autoantibodies were detected in the urine but not in the paired serum samples, in contrast with the results reported by Meryhew et al. [33], where the specific antibodies detected in urine were always present in the corresponding serum sample. According to the authors, this difference was attributed to the greater number of patients examined in the former study [35].

**Table 1 ijms-22-07143-t001:** Summary of immune-related urinary biomarkers of LN.

Urinary Biomarker Class	Diagnostic Value	Prognostic Utility	Response to Treatment
Autoantibodies	Anti-RNAPI, anti-dsDNA, anti-La, and anti-ribosomal P, levels correlated with disease activity [35]		
FLC	FLC discriminate patients with severe forms of LN [36,37]	FLC increase before the onset of acute SLE relapses and reach normal values after remission [38,39,40]	λ and κ FLC decrease after treatment [37]
Complement components	C3d levels correlate with SLEDAI discriminate between active LN and inactive LN or non-renal SLE [41,42,43]	C3d decreased levels can predict treatment response at 6 months and non-response or flare [43]	C3d levels fall after therapy [43]
Soluble immune mediators	IL-6 higher in active LN [44] corroborated by renal biopsy [45]	No differences between active or inactive LN [46].	Decreased significantly after treatment [47]
	MCP-1 correlates with LN activity [48]. Higher in patients with inactive LN (Meta-analysis) [49]. Increased also in serum of SLE patients [50] MCP-1, KIM-1, and NGAL higher in patients with active LN [51]		
	IP-10 positively correlated with renal SLEDAI but not significantly higher in LN [52]		
	EGF lower in patients with active LN [53]	Decreased overtime in adverse long-term kidney damage [53]	
	VCAM-1 higher in active renal disease [54]. Presence of LN, clinical and histological activity indexes severe renal lesions [55,56] VCAM-1 and ALCAM elevated in active LN [57,58] ICAM-1 elevated in SLE patients [59] VCAM-1, cystatin C, and KIM-1 discrimination between proliferative versus membranous LN. Non-specific for SLE [60] NGAL; higher in LN than in non-LN patients [61]	Increased in active LN [56]. It may indicate patients at increased risk for long-term renal function loss [57]. ALCAM levels correlated positively with activity index [58]	Effective LN therapy reduced uVCAM-1 levels over the time [56]
Leukocytes	Monocytes/macrophages in proliferative LN [62,63,64] Higher eosinophils numbers in LN [65] CD4+CD25-Foxp3+ T cells in active LN [66] CD4+ and CD8+ T cells in active LN [67,68] Th17 associated with less severe disease [69] pDC and PB/PC in severe LN [70,71]		Lower numbers of CD8+ T cells in remission [67,68]
Soluble leukocyte marker	sCD163 in active LN [72,73], mostly in proliferative classes [74] sCD11b correlates with histological activity [75] T-bet mRNA in higher in active LN [76,77]	sCD163 increases from 6 months before flare [78] Higher T-bet mRNA gives higher risk of severe flare [77]	sCD163 decreases after treatment in drug responders [73,78] sCD11b decreases with clinical remission [75]

However, other studies failed to detect anti-dsDNA antibodies. Yamada et al. [79], studied the presence of anti-dsDNA in serum and urine of nine SLE patients presenting heavy proteinuria. While they were able to detect anti-dsDNA in serum of these patients, they were not found in urine. The same result was reported by Pérez-Vázquez et al. [80]. Later, Macanovic et al. [81], proposed the inclusion of EDTA in the immunoassay to allow the detection of anti-dsDNA antibodies in the urine of LN patients with proteinuria, but not in SLE patients without renal involvement. The decrease in autoantibody titers in serum of LN patients observed in these studies may suggest their deposition in the kidney, a process that most likely involves their interaction with the autoantigen to form IC, and an increase in disease severity. This hypothesis agrees with the finding of antibodies directed against the light chain of laminin in the extracellular matrix, mainly in the urine of active LN patients, which was in a much lower concentration in the matched serum samples. These autoantibodies might cross-react with anti-dsDNA autoantibodies [82].

Apart from anti-nuclear autoantibodies, few reports have been carried out to study the presence of autoantibodies of other specificities in urine of LN patients. It is well known that patients with SLE often exhibit an aberrant interferon (IFN) response. A gene profiling study has revealed increased levels of IFN-stimulated genes in blood and tissues of SLE patients [83,84]. Increased IFN activity has been associated with renal disease [85], so, due to the importance of IFN in SLE pathogenesis, measurement of IFN activity may be useful for guiding therapy decisions. However, IFN activity measurements may be influenced by the production of anti-IFN autoantibodies. It has been described that, in SLE patients, anti-IFN autoantibodies have been associated with blockade of IFN signaling and lower disease activity [1]. As IFN activity is associated with kidney damage in SLE, Harris et al. [86], examined the relationship of IFN activity and anti-IFNα autoantibodies in serum and urine of SLE patients. They found that IFN activity was higher in the urine of SLE patients compared to healthy donors, but it did not find a correlation between urine and serum samples. Interestingly, the analysis of anti-IFNα autoantibodies revealed an inverse correlation between blood and urine concentrations. Autoantibodies against both IFNα1 and IFNα6 were observed in urine, but they were rarely observed in serum. These differences may suggest that local immune responses in the kidney would be distinct from those detected in the serum.

In summary, although there are very few studies reporting the clinical significance of urine autoantibodies, the analysis of their distinct specificities in urine and serum has the potential to become a useful tool for the diagnosis and monitoring of the renal disease activity in SLE patients. Nevertheless, this should be confirmed in larger and serial sampling studies.

## 3. Free Light Chains

B cell activation has an important role in the pathogenesis of SLE. It is well known that during the flares an extensive polyclonal B cell hyperactivity is observed, followed by an exacerbated synthesis of immunoglobulins [87]. Human immunoglobulins produced by B cells are made of two heavy chains and two free light chains (FLC): κ and λ. Under normal conditions, the production of antibodies is accompanied by an excess of FLC over heavy chains, which are secreted to the circulation, filtered by renal glomeruli, and reabsorbed in the proximal tube (Figure 1). Thus, they are only found in very low levels in the urine in healthy individuals. However, in chronic inflammatory diseases such as SLE, elevated levels of urine FLC can be found as a result of increased production overcoming the renal clearance capacity or renal tubule impairment [88,89].

Currently, the study of FLC is an area of growing interest in order to better understand their biological role and their clinical use in different chronic inflammatory and autoimmune diseases [90]. Several reports have been published in last years about the potential use of FLC as biomarkers of disease activity in different autoimmune pathologies. In fact, some analyses have shown an association between serum FLC levels and disease activity in SLE and RA patients [88,91,92]. Furthermore, in Sjogren’s syndrome, the presence of immunological activity markers and extraglandular manifestations were correlated with elevated levels of serum FLC [38,93]. However, few studies have evaluated the levels of FLC in urine of patients with autoimmune diseases.

The increase in FLC in the urine of patients with SLE has been described during active renal disease in the 70s [94], suggesting that quantitative alterations in urinary FLC excretion might represent an early predictor of disease relapse and remission. Later studies reported a significant increase in urinary FLC 4–8 weeks prior to the onset of acute SLE relapses, suggesting that a time frame may exist between the occurrence of immunopathologic B-cell stimulation and the resultant symptoms and tissue damage mediated by IC-induced acute inflammation [39]. Urinary FLC were increased during active phases of SLE, whereas they reached normal values after remission, supporting the hypothesis that levels of urine FLC may be used to monitor disease-related B-cell activity in SLE [40]. Similar to increased levels of soluble interleukin-2 receptors, FLC and the presence of cytokine-like molecules in urine can directly reflect the severity of inflammatory and immunological reactions in patients with LN [36] (See Table 1).

A study comparing serum and urinary FLC in SLE patients revealed high serum FLC levels with a simultaneous increase in FLC in urine. However, some patients presented high urinary FLC in the absence of detectable serum FLC. There was no correlation between serum or urinary FLC and occurrence or severity of renal disease [95]. This result is in agreement with a more recent study showing no correlation between markers of systemic inflammation and FLC levels [88]. The investigation of both serum and urinary FLC in LN patients with class III/IV and non-class III/IV showed higher levels of FLC in class III/IV, suggesting that urinary FLC levels may be useful as biomarkers for the detection of more active or severe forms of LN. Serum and urinary FLC did not correlate with disease activity, anti-dsDNA and low complement. However, levels of FLC in urine showed a strong correlation with proteinuria and plasma cell infiltration of the kidney. Furthermore, both urinary and serum FLC were lower after treatment, thus providing evidence for a possible direct biomarker of renal inflammations and local pathogenesis [37] (Table 1). This study presented some limitations, as they lacked a control group and the sample size was small. The authors were unable to confirm whether the higher urinary FLC levels observed in class III/IV LN was secondary to the increase in proteinuria, or that the decrease in urinary FLC post-treatment was also due to the decrease in general proteinuria [96].

Urine FLC were also used to differentiate between active LN and secondary infections, as these two conditions are often associated, and they might not be distinguished easily. There were differences in the urinary levels of FLC between healthy subjects and LN patients. Nevertheless, this analyte was unable to distinguish patients with infection from those without it [87].

Despite the fact that it is still unknown how FLC are mediating their biological functions, it is clear that they play an important role in the pathogenesis of SLE with renal damage. All these studies showed significant findings in the field of urinary FLC as biomarkers for LN. However, urinary FLC are not specific for LN, as they may be also increasing in other glomerulonephritis characterized by plasma cell infiltration [37]. Therefore, more studies of FLC in urine may help to understand their role in different diseases, and provide a useful parameter to monitor disease progression. Furthermore, larger longitudinal studies are needed to determine the predictive value of urinary FLC as biomarkers of disease activity and relapse, as well as treatment response.

## 4. Complement

The complement is part of the innate immune system and is one of the main effector mechanisms of antibody-mediated immunity. The complement system is composed by more than 30 plasma and membrane-bound proteins that function as a cascade. There are three pathways of complement activation: classical, alternative, and lectin pathway [97]. While the involvement of the complement in the pathogenesis of SLE is well accepted, its exact role is still not clear. On the one hand, complement components appear to mediate autoantibody-initiated tissue damage. (Figure 1). Hereditary deficiencies of complement components are associated with an increased risk for the disease [98]. The deposition in the kidney of IC formed by autoantibodies directed against a vast range of self-antigens, and the subsequent activation of the classical pathway are considered major mechanisms mediating tissue injury in LN [97]. Moreover, the involvement of the alternative or lectin pathway has also been suggested in several studies [99,100,101,102,103].

It has long been recognized that serum C3 and C4 levels generally are lower in SLE patients [104]. However, serum low complement levels have proven disappointing as disease activity markers in SLE due to their persistency at low or normal levels, independently of disease activity, and their low sensitivity at predicting flares [97]. Based on the belief that complement split products reflect the activation of the complement more accurately than the levels of the individual intact proteins, several studies have explored their potential use as blood biomarkers of disease activity. These include levels of complement products such as C3a, C4a, C5a, iC3b, C3d, C4d, Ba, Bb, and soluble C5b-9 [104,105]. However, there are technical issues related to detection of these split products in plasma (very short life-term, or activation in vitro at room temperature) that have made these tests impractical for general clinical use outside of research purposes [106]. As there has been no agreement in the use of soluble complement products as blood biomarkers, urine detection still is an attractive option.

It has been described that complement components can be found in the urine of LN patients, particularly in patients with active kidney disease, and they may be indicative of complement activation in kidney and reflect renal inflammation. These studies have reported the presence of C3 fragments, Ba, Bb, C4d, C5a, and C5b-9 in the urine of patients with LN [41,107,108,109,110]. Factor H has also been detected in the urine of LN patients [110,111], but the most studied has been urine levels of fragmented C3d. Detection of urinary excretion of C3d fragments among SLE patients using Western blot was variable among SLE patients with non-renal manifestations [109]. Other studies using a reduced number of patients also found C3d in urine of patients with LN, but also found them in patients with no evidence of renal disease or proteinuria, suggesting a non-renal origin of C3d (Table 1). Nevertheless, urine C3d levels were better than plasma C3, C4, C4d, C5b-9, and anti-dsDNA to differentiate acute from chronic LN [41]. These results agree with a later study reporting elevated urine C3d only in patients with active LN compared to inactive LN and non-renal SLE. Moreover, urine C3d showed a stronger correlation with the Systemic Lupus Erythematosus Disease Activity Index (SLEDAI) than serum C3, C3d, C4, or anti-dsDNA antibodies (Table 1). The authors hypothesized that C3d excretion was partly the result of local production of C3d in the kidneys, as they were unable to find correlations between urine and plasma C3d as well as the C3d/C3 ratio [42]. Finally, a longitudinal study measured levels of C3d before and after 3 months of induction therapy in a cohort of biopsy-proven LN patients. LN patients showed high levels of urinary C3d at onset, and a significant fall at the end of treatment in the entire cohort. In addition, levels of urinary C3d correlated with SLEDAI, but there was no significant correlation with plasma C3 and C4, suggesting an intra-renal production of C3d (Table 1). Urinary C3d/creatinine values could discriminate between active and inactive nephritis [43].

In summary, complement fragments indicative of complement activation can be found in the urine of LN patients, particularly in patients with active kidney disease. These studies support the idea that urinary C3d levels correlate more tightly than other markers with LN disease activity, so that they can be used as biomarkers to determine response to treatment, identify non-responders or relapses. However, larger studies involving a greater number of patients and controls, as well as different treatment modalities, are required to further validate the use of urinary C3d as a biomarker of LN.

## 5. Soluble Immune Mediators

Most of the soluble immune-related molecules present in urine includes cytokines and chemokines, growth factors of others (Figure 1). They are easy to detect and normally their level changes are indicative of inflammation. Herein, we will focus on the inflammatory mediators present in urine with potential use as biomarkers [3,22].

In the kidney, cytokines and chemokines are produced locally by infiltrating inflammatory cells. One of the most studied proinflammatory cytokine in SLE patients is IL-6. This cytokine is produced in response to inflammatory stimuli, having a key role in autoimmunity [112]. The initial SLE studies were focused on the evaluation of IL-6 levels in the serum of SLE patients [113]. Later, the increase level of IL-6 in urine was described as a potential marker to follow the disease. Levels of IL-6 were evaluated, resulting higher in patients with LN, class IV. Additionally, IL-6 levels decreased significantly after treatment [47]. A posterior study described the increased detection of IL-6 in the tubules and glomeruli of LN kidneys [114]. More recently, level of IL-6 in urine was measured together with β2 microglobulin (b2M), Tamm–Horsfall glycoprotein, IL-2Rc, and IL-8. Levels of IL-6 and IL-8 were higher in the urine of patients with active LN, compared to those with inactive LN and normal individuals. The levels of b2M were significantly elevated in patients with LN compared to controls. However, no significant differences were detected in the level of b2M between active or inactive LN [44]. In a more recent study, the levels of IL-6 were evaluated in urine and serum. In this case, renal biopsies were also performed prior to or shortly after urine and serum sampling. No differences were observed in IL-6 serum levels, although urinary levels of IL-6 were higher in SLE patients with LN, corroborated by biopsy [45]. Alternative approaches have been focused on the study of markers indicative of oxidative stress including malonyldialdehyde, oxidized-to-total glutathione, catalase, and superoxide dismutase (SOD). In a total of 65 SLE patients, urine was analyzed for the presence of oxidative stress markers, monocyte chemoattractant protein-1 (MCP-1), IL-6, and C-reactive protein (CRP). The study also included the detection of classical markers, as the total urine protein/creatinine ratio, serum anti-dsDNA, anti-C1q antibodies, and complement proteins C3 and C4. All these biomarkers showed significant differences between active and inactive LN patients, and between active LN and non-LN with the exception of IL-6, SOD, and catalase [46].

Several chemokines have also been measured in the urine of LN patients (Table 1). One of them is MCP-1, also known as CCL2. This molecule is involved in the migration and infiltration processes of monocytes, having a key role in tissue injury. MCP-1 was shown to be associated with the proliferative reaction in autoimmune diseases, and several studies have demonstrated that it can be upregulated during the inflammation process. The role of MCP-1 in LN has been evaluated by several groups [115,116]. Urine levels of MCP-1 have been proposed to correlate with LN activity [48]. In a recent meta-analysis including eight different studies and a total of 399 patients, the urinary level of MCP-1 was higher when comparing active LN with inactive LN patients, or with healthy controls. The meta-analysis also corroborated that the level of MCP-1 was higher in patients with inactive LN than in healthy controls [49].

Other studies have evaluated the levels of MCP-1 together with other chemokines. Ding et al., included the determination of MCP-1 together with kidney injury molecule-1 (KIM-1) and neutrophil gelatinase-associated lipocalin (NGAL) in urine. Levels of these three molecules were higher in patients with active nephritis, compared to LN patients in remission and normal controls, and higher in patients with active tubulointerstitial lesions compared to those with chronic lesions [51]. MCP-1 has also been evaluated in combination with other potential biomarkers, such as TNF-like weak inducer of apoptosis (TWEAK) and NGAL, in the serum and urine of SLE andanti-neutrophil cytoplasmic antibody (ANCA) associated vasculitis (AAV) patients. Although levels of all biomarkers were correlated with SLEDAI, there were no differences between active LN and active renal AAV [117]. Based on these results, the increased level of these biomarkers could be non-specific of LN, and determination of these cytokines for diagnosis must be taken with caution. Additionally, in a recent study, correlation between MCP-1 levels in serum and urine has been studied in a cohort of patients with SLE with or without LN and controls. A significant increase in MCP-1 was detected in serum and urine of all SLE patients [50]. In this line, in a study including a total of 87 patients with SLE, MCP-1 was measured in urine using the FIDIS multiplex bead assay, together with normal T cells expressed and secreted [118], soluble tumor necrosis factor receptor 1 (sTNF-R1), interferon-inducible protein 10 (IP-10), monocyte inhibitory protein 1α (MIP-1α), and vascular endothelial growth factor (VEGF). Significant differences were found between active LN and non-renal SLE for VEGF, and levels of sTNF-R1 and IP-10 in urine and serum correlated with SLEDAI scores. Based on this data, the sTNF-R1 and VEGF could be used as markers of disease activity in SLE and LN. The relationship between SLE and IP-10 serum level has also been studied by several groups. The serum level of IP-10 was not different between active and non-active LN, but was positively correlated with SLEDAI (Table 1). The urine level of IP-10 showed a non-significant trend to be higher in patients with active LN, even though IP-10 level showed a positive correlation with renal SLEDAI [52].

B-cell-activating factor of the tumor necrosis factor family (BAFF) has also been quantified by ELISA in urine from patients with different autoimmune diseases. BAFF is involved in the survival and maturation process of peripheral B cells, as well as in T and B cell activation. In a recent study including patients with different autoimmune diseases (SLE, Sjögren syndrome, IgA nephropathy, and healthy controls), urine levels of BAFF were higher in patients with SLE and in patients with active renal disease [119].

Urinary Epidermal Growth Factor (EGF) has also been evaluated as a possible LN biomarker (Table 1). The urinary EGF levels in LN patients were determined in a longitudinal study. Its level was lower in patients with active LN compared to patients with active non-renal SLE, patients with inactive SLE and healthy kidney donors. In addition, the urinary EGF level was inversely correlated with the chronicity index assessed by kidney biopsy histology. In a follow up of the patients, it was corroborated that urinary EGF were lower at flare and were decreasing over time in the case of adverse long-term kidney damage [53].

Another urinary biomarker that has been widely studied in LN is the vascular cell adhesion molecule 1 (VCAM-1) (Table 1). This protein is also known as CD106, and is involved in the progression of glomerular and tubulointerstitial injury in LN. In a cohort of 227 SLE patients (80 inactive SLE, 67 active non-renal, and 80 active renal disease) and 53 controls, VCAM-1 was significantly higher in urine from patients with active renal disease compared to patients with active non-renal disease. In this study, the urinary levels of CXCL1 and angiostatin were also increased in patients with active renal disease, and they were correlated with total SLEDAI and renal SLEDAI scores, and with the urinary protein/creatinine ratio [54].

The soluble urine VCAM-1 showed a strong association with the presence of LN, with clinical and histological activity indexes and with more severe renal lesions [55,56]. Additionally, in a prospective study with class II, IV, and V LN patients classified as active or inactive nephritis at the inclusion time; evaluation of the urinary VCAM-1 showed increased levels in active LN. In this study, they also detected an increased level of VCAM-1 in serum [56].

Another recent study has shown increased levels of VCAM-1 in urine from SLE patients. VCAM-1 and activated leucocyte cell adhesion molecule (ALCAM) were determined in a cohort of active and inactive LN patients. Urine levels of VCAM-1 and ALCAM were elevated in patients with active LN compared to healthy controls and with quiescent nephritis. A positive correlation was established between urine ALCAM and SLEDAI. Levels of VCAM-1 and ALCAM were elevated in patients with active LN, and the ALCAM level was higher in proliferative the classes III and IV. Of both, higher levels of VCAM-1 in urine could be indicative of a long-term renal function loss, while ALCAM could be used as a potential biomarker of kidney disease [57,58].

Another molecule with a relevant role in LN is the intercellular cell adhesion molecule ICAM-1. It is expressed in leukocytes and endothelial cells, and is involved in binding and transmigration processes after leukocyte activation. In a recent systematic review searching for studies that compare blood and/or urinary ICAM-1 in SLE patients, a total of 1215 articles were found. From those, 22 articles were included in a meta-analysis. The analysis of seven studies comparing urinary ICAM-1 in SLE patients versus healthy controls revealed that ICAM-1 level was higher in SLE patients (Table 1). Similarly, ICAM-1 was increased in the blood, but did not allow the discrimination between active and inactive SLE [59].

Using high-throughput proteomic technologies, the expression levels of VCAM-1 and nine other urinary biomarkers (b2M, calbindin D, cystatin C, IL-18, KIM-1, MCP-1, nephrin, NGAL, and Vitamin D binding protein (VDBP)) were tested in SLE patients with active or inactive nephritis. All these markers, except nephrin, were significantly increased in active LN compared to healthy controls. Cystatin C, MCP-1, and KIM-1 levels were significantly higher in active LN group compared to inactive LN group and a positive correlation was established between cystatin C, KIM-1, MCP-1, NGAL, VDBP, and the renal SLEDAI. The values of VCAM-1, cystatin C, and KIM-1 allowed the researchers to discriminate between proliferative versus membranous LN. However, these markers are not specific for SLE, and they can be increased in urine due to other kidney-affecting diseases [60].

Finally, it is worth mentioning other immune-related enzymes or molecules present in urine from SLE patients that could be used as biomarkers. One example is the NGAL, expressed in neutrophils. This molecule is involved in the iron trafficking in the kidney and in the inflammatory responses by sequestering neutrophil chemoattractants [120]. NGAL expression is induced in the nephron in response to epithelial injury [121] and its level appeared increased in urine after renal injury [122,123]. In a study including a total of 70 SLE patients and 20 controls, the levels of urinary NGAL in LN patients were significantly higher than in non-LN patients [61].

In the last years, new technological advances have allowed to increase the number of molecules that can be detected in urine simultaneously. One example is the use of arrays. In a recent study, a screening in urine for 1000 different proteins (including proteins with immune and non-immune functions) has been performed in order to distinguish between active and inactive renal disease. The study was performed using a capture antibody-coated protein array. The resulting candidates were corroborated by single cell analysis. A total of 64 proteins were identified in the urine of SLE patients, from which 17 were validated in independent cohorts. The level in urine of angiopoetin-like 4 (Angptl4), L-selectin, tripeptidyl peptidase 1 (TPPI), transforming growth factor-β1 (TGFβ1), thrombospondin-1, folate receptor beta (FOLR2), platelet-derived growth factor receptor-β, and peroxiredoxin 2 (PRX2) allowed to differentiate between active and inactive renal disease. Multivariate regression analysis showed a high association of urine Angptl4, L-selectin, TPPI, and TGFβ1 with disease activity [124].

In conclusion, there are several cytokines, chemokines, and other soluble immune mediators present in urine that could be used as biomarkers of kidney damage in SLE. However, we still do not have a definitive panel of molecules allowing a precise diagnosis of renal damage in SLE, or to differentiate the stages of renal disease.

## 6. Cell-Associated Biomarkers

Urine leukocytes and erythrocytes are often present in the urine of LN patients (Figure 1). However, it is only recently that the identification of immune cell populations has gained interest as a tool to estimate kidney infiltration. Urine cells can be collected by urine centrifugation and analyzed by flow cytometry or RNAseq. The resulting cell profiles can be compared to those of kidney biopsies, or alternatively correlated with the activity of the disease or drug responses. Leukocyte infiltration monitoring using urine is simple and non-invasive, can be repeated for serial assessments, can be analyzed rapidly and inexpensively, and provides a high-dimensional panorama of the local infiltrates. However, it presents some limitations, as some cells could die rapidly in the urine and only cells from specific regions of the kidney can void in the urine, as reviewed in [21]. Interestingly, a recent study based in single cell RNAseq (scRNAseq) on healthy donors demonstrated that almost all kidney cell types can be identified in the urine, in addition to myeloid cells and lymphocytes [19]. Moreover, a comparison of the populations in the urine sediment with those infiltrating the kidney of LN patients using scRNAseq evidenced a high correlation between the transcriptional signatures of the different populations in both tissues [70]. Thus, at this regard, urine leukocytes can be potentially used as a surrogate for leukocyte infiltration in the kidney biopsies.

Mass cytometry deep immune phenotyping performed in the urine of active LN patients has shown that the most prominent cell populations are neutrophils and monocytes/macrophages and, in a lower extent, T and B lymphocytes, eosinophils, and natural killer cells [125]. CD14^+^/CD16^+^ monocytes/macrophages are detected not only in the urine of LN patients, but also in other proliferative glomerulonephritis (Table 1). These cells were more abundant during the acute exacerbation of renal disease, while they disappeared during remission [126]. Other authors have also described CD14^+^ cells as a major population in the urine of LN. CD14^+^ cells were more abundant in LN compared to non-renal SLE and healthy controls, and associated with class IV but not in class III nephritis [62]. More precisely, Kim et al. [63], found higher numbers of CD11c^+^ macrophages in the urine of patients with proliferative LN, with a strong association with the serum anti-dsDNA titers and chronicity indexes. They were also associated to a poor renal response to the treatment with immunosuppresants. Their surface receptor expression pointed to infiltrated monocytes rather than kidney-resident macrophages, and produced pro-inflammatory cytokines, such as TNFα, IL-6, and IL-1β. They showed a high expression of CXCR3, which was correlated with IP-10 expression in urinary tubular epithelial cells [64].

The finding of monocytes/macrophages in urine showed a good correlation with several descriptions of infiltrates of monocytes/macrophages in kidney biopsies [127]. Cell imputation, based on gene expression profiles in the kidney biopsies of LN patients and live kidney donors using CIBERSORT, showed that the monocyte signature is associated with nephritis [128]. Histological analysis of kidney biopsies of LN patients evidenced an accumulation of glomerular patrolling monocytes, related to vascular inflammation in class IV-G LN patients, characterized as CD16^+^ CD14^−^ CD15^−^ [129]. Another study reported that CD16^+^ monocytes infiltrating the glomeruli of active LN patients express high levels of the chemokine fractalkine, correlating with markers of activity, such as low levels of circulating complement, anti-dsDNA antibodies, proteinuria, or glomerular filtration rate (GFR) [130]. Infiltrating CD16^+^ cells were negatively correlated to a reduction in circulating CD14^+^ CD16^++^ monocytes in severe forms of LN [131], indicating that these cells are actively migrating to the kidney in active LN patients.

CD163 is an endocytic receptor of haptoglobin-hemoglobin complexes, expressed in the membrane of M2 macrophages. A soluble form of the protein (sCD163) with low binding affinity is shed to the extracellular matrix upon macrophage stimulation, and can be detected in the serum as a marker of macrophage activity [132]. Immunohistological analysis of the glomeruli of LN patients have shown that a high proportion of infiltrating macrophages express CD163 in the glomeruli [133], and the extent of CD163 expression was associated with renal severity (Table 1). Urinary sCD163 was strongly correlated with renal CD163 and histological scores [72]. Moreover, urine sCD163 increases from 6 months before the renal flare, and perfectly agreed with the histological activity index in repeated biopsies. Additionally, a combination of proteinuria and urinary sCD163 improved the prediction of those LN patients that would achieve complete renal responses at 12 months [78]. Another longitudinal study including 122 SLE patients measured paired plasma/urine sCD163 levels, showing much higher urine sCD163 levels in active LN. After the treatment, sCD163 decreased more consistently in urine than in plasma, performing better than plasma complement or anti-dsDNA titers for the differentiation between LN and non-renal SLE patients (Table 1). In the patients experiencing a renal relapse, urinary sCD163 concentration correlated with the changes in disease activity [73]. A trans-ethnic study evidenced high urine sCD163 in LN patients of African American, Caucasian, and Asian LN subjects. Urine sCD163 was associated with proliferative histological classes and correlated with renal SLEDAI, but not with chronicity parameters, demonstrating again its potential as a tool to predict renal pathology [74].

CD11b is expressed in the surface of neutrophils and macrophages. It supports various immunological functions in the glomerular endothelium after IC deposition, including leukocyte recruitment and IC clearance [134,135]. A robust association between a functional variant of CD11b (ITGAM) and a risk of developing SLE has been described [136]. Of note, CD11b can also be proteolyzed and released to the extracellular milieu under inflammatory conditions [135]. Urinary levels of sCD11b were correlated with the number of glomerular leukocytes and the histological activity (Table 1). Moreover, sCD11b performed better than sCD163 for the prediction of LN. Finally, urinary sCD11b also decreased after successful glucocorticoid treatment, and performed better than sCD163 for the prediction of LN [75]. Thus, urine sCD11b can be very useful for the monitoring of LN activity and therapeutic failure, as well as for the detection of glomerular leukocyte accumulation.

As mentioned before, granulocytes are found in the kidney and in the urine of LN patients. They are mainly neutrophils and, in a lesser extent, eosinophils [125]. In addition, eosinophils have also been described as enriched in the urine of LN patients, compared to non-nephritis SLE. This finding was associated with higher urinary concentrations of eosinophil cationic protein (EPC) and IL-5, that could then be surrogate markers of eosinophiluria [65].

Regarding T lymphocytes, Chan et al., reported a higher number of CD3^+^ cells in active LN, which correlated with SLEDAI and renal scores [76]. In a paired urine/blood cytometry study focusing on the quantification of macrophages and B and T lymphocytes in several immune-mediated nephropathies, the best ROC curves were obtained using urine CD4^+^ and CD8^+^ T cell counts for the detection of renal damage in SLE patients, being even better than high creatinine detection. Moreover, urinary CD4^+^ > 800/100 mL were found exclusively in active LN patients, and their numbers normalized after treatment [67]. Urinary CD4^+^ T cells showed mainly an effector memory phenotype, and regulatory T cells (Treg) counts correlated with disease activity [137]. Other authors have also proposed a threshold of uCD4^+^ > 800/100 mL to detect active LN patients, which were persistent in proliferative nephritis with a worse outcome. Urine CD8^+^ T cells also showed a memory phenotype in active LN patients [68], and were also detectable in LN biopsies [138]. More precisely, type 1 helper (Th1) marker T-bet mRNA was found in active LN urines and tubular expression of T-bet was associated with histological activity [77] and predicted severe flare in a longitudinal study [77]. Accordingly, it has been described an inverse association between Th1 cells in the blood and in the urinary/renal tissue, indicating a role of Th1 cells in the physiopathology of LN. Urinary type 17 helper (Th17) were associated with a less severe nephritis, and they were increased in blood after treatment [69]. Concerning Treg, a population of urinary CD4^+^ CD25^−^ Foxp3^+^ T cells showed a positive correlation with proteinuria in active LN patients (Table 1). Although they seemed to be functional Treg, they showed lower suppressor activity than classical Treg [66]. Foxp3 expression was higher in the kidney of SLE patients with proliferative nephritis, being associated with higher severity and activity [139].

Kidney-infiltrating B cells is a common finding in LN biopsies [70]. However, urine B cells have been described in LN patients in very low quantities [125], with a phenotype of Ig-secreting plasmablasts or plasma cells (PB/PC) and associated with proliferative nephritis. In addition, plasmacytoid dendritic cells (pDC) often accompanied B cell in the urine, associated with detectable urinary IFNα/β [71] (Table 1). Interestingly, a clear IFN response could be observed in most infiltrating cells using scRNAseq, which correlated with gene expression modules in the urine [70]. A histological analysis revealed an infiltrate of conventional dendritic cells (cDC) and pDC with an immature phenotype in class III/IV LN patients [140], indicating a role in the immunopathogenesis of LN [127].

## 7. Concluding Remarks

Urine is gaining interest as a non-invasive source of information about the inflammatory status of the kidney in SLE patients. All the immune mediators of LN pathogenesis have been detected in the urine of LN patients, and in most of the cases it has been shown their correlation with severity, activity, or response to treatment. Nowadays, high content technologies, such as multiplexed immune assays, transcriptomics, proteomics, or mass cytometry, are giving a new boost to this field, allowing a discovery strategy for the identification of promising molecules or populations. Therefore, it is likely that new biomarkers will arrive in a near future, with a high positive impact in the quality of life of LN patients.

## Figures and Tables

**Figure 1 ijms-22-07143-f001:**
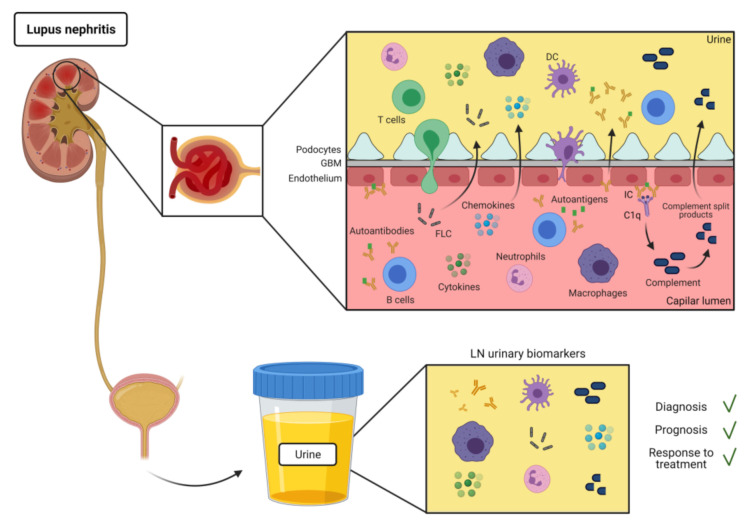
Physiopathology of LN and urine biomarkers. Renal damage in LN is mediated by the infiltration of effector leukocytes, autoantibody binding to nuclear and non-nuclear autoantigens, and generation of IC. These IC are deposed in the glomeruli, affecting the kidney function and leading to an inflammatory cascade. Consequently, filtering of the blood is hindered, and many immune-related cells and molecules involved in the inflammatory response may be excreted into urine. Assessment of these molecules in urine may help to predict the development of LN, renal flares, as well as response to treatment. GBM: Glomerular basement membrane; IC: immune complex; DC: dendritic cell; FLC free light chains.

## Data Availability

Not applicable.

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
