# Peer review of "Immune-Related Urine Biomarkers for the Diagnosis of Lupus Nephritis"

_ijms, 2021, doi:10.3390/ijms22137143_

Round 1

Reviewer 1 Report

Very inetresting review, concisely written, on clinically important topic.

Minor suggestions:

  1. Due to a wide variety of biomarkers described in this papaer, it would be more comprehensive for the readers to summarize them in a table (for example presenting their clinical associations / significance).
  2. At the end of the text it would be interesting to read concluding remarks and authors comment on the perspectives related to the bimarkers development.
  3. Spelling required. For example: repeated phrase (line 89); explanation of abbreviations first used in the text (section 2. Autoantibodies); growth factors (line 288); focus in evaluate (line 294), etc.

Author Response

Thank you for your positive comments. We have addressed your suggestions as follows:

  1. We have included a table (table 1) summarizing the information of the most relevant biomarkers in function of their clinical relevance
  2. We have included a “concluding remarks” section
  3. We have corrected the spelling issues you have identified. Additionally, we have performed a general editing to the text in order to ameliorate its readability

Reviewer 2 Report

The objectives and novelty of the article should be specified compared to existing ones, for example https://pubmed.ncbi.nlm.nih.gov/32743523/;

- I think it would be useful a table structured on the studies found or on markers and the main objective results (diagnostic value, prognosis)

- “Despite the fact that there are very few studies reporting the clinical significance of 161 urine autoantibodies, the existing data suggest that the analysis of autoantibodies of dis-162 tinct specificities in urine and serum may have value for diagnosis and monitoring of the 163 renal disease activity in SLE patients. ” - I think it should be slightly reformulated, because it contradicts the above data, some studies have not observed any association;

- I think it would be useful a section with conclusions at the end

-the phrase “Under normal conditions, the production of antibodies is accompanied by an excess of FLC over heavy chains. Which are secreted to the circulation, filtered by renal glomeruli and reabsorbed in the proximal tube. ”

Author Response

We want to thank the reviewer for his/her critical reading. We feel that the suggestions and comments helped us to submit a better manuscript.

1.- The field of urine biomarkers in lupus nephritis is evolving rapidly, and thus other excellent reviews have been published recently, as Aragón et al, 2019. In this manuscript we have put the focus in the immune effectors present in the kidney of LN patients to give an integrative perspective of both tissue pathogenesis and urine biomarkers. Using this perspective, we have gone beyond other publications, since in addition of soluble cytokines/chemokines or mRNA, we have extended the topic to the detection of complement and antibody components, autoantibodies and a wide range of effector cells. In our knowledge it is the first publication that includes these layers of information.

2.- We have included a table (table 1) summarizing the information of the most relevant biomarkers in function of their clinical relevance

3.- We agree with the reviewer that the message of this sentence can be in contradiction with the statements of the section. We have corrected it.

3.- We have corrected the sentences. Additionally, we have performed a general editing to the text in order to ameliorate its readability

Round 2

Reviewer 2 Report

No further questions.
Thank you for the improvement.